# REVISITING GRAPH NEURAL NETWORKS FOR LINK PREDICTION

## ABSTRACT

Graph neural networks (GNNs) have achieved great success in recent years. Three most common applications include node classification, link prediction, and graph classification. While there is rich literature on node classification and graph classification, GNNs for link prediction is relatively less studied and less understood. Two representative classes of methods exist: GAE and SEAL. GAE (Graph Autoencoder) first uses a GNN to learn node embeddings for all nodes, and then aggregates the embeddings of the source and target nodes as their link representation. SEAL extracts a subgraph around the source and target nodes, labels the nodes in the subgraph, and then uses a GNN to learn a link representation from the labeled subgraph. In this paper, we thoroughly discuss the differences between these two classes of methods, and conclude that simply aggregating *node* embeddings does not lead to effective *link* representations, while learning from *properly labeled subgraphs* around links provides highly expressive and generalizable link representations. Experiments on the recent large-scale OGB link prediction datasets show that SEAL has up to 195% performance gains over GAE methods, achieving new state-of-the-art results on 3 out of 4 datasets.

## 1 INTRODUCTION

Link prediction is to predict potential or missing links connecting pairwise nodes in a network. It has wide applications in various fields, such as friend recommendation in social networks (Adamic & Adar, 2003), movie recommendation in Netflix (Bennett et al., 2007), protein-protein interaction prediction (Qi et al., 2006), and knowledge graph completion (Nickel et al., 2015), etc.

Traditional link prediction approaches include heuristic methods, embedding methods, and feature-based methods. Heuristic methods compute some heuristic node similarity scores as the likelihood of links (Liben-Nowell & Kleinberg, 2007), such as common neighbors, preferential attachment (Barabási & Albert, 1999), and Katz index (Katz, 1953), which can be regarded as some predefined graph structure features. Embedding methods, including matrix factorization (MF) and Node2vec (Grover & Leskovec, 2016), learn free-parameter node embeddings from the observed network transductively, thus do not generalize to unseen nodes and networks. Feature-based methods only use explicit node features yet do not consider the graph structure. Recently, graph neural networks (GNNs) emerged to be powerful tools for learning over graph-structured data (Scarselli et al., 2009; Bruna et al., 2013; Duvenaud et al., 2015; Li et al., 2015; Kipf & Welling, 2016a; Niepert et al., 2016; Dai et al., 2016), and have been successfully used in link prediction as well (Kipf & Welling, 2016b; Zhang & Chen, 2018; You et al., 2019; Chami et al., 2019; Li et al., 2020).

There are two main types of GNN-based link prediction methods. One is Graph Autoencoder (Kipf & Welling, 2016b), where a GNN is first applied to the entire network to learn an embedding vector for each node. Then the embeddings of the source and target nodes are aggregated to predict the target link. The second type is SEAL (Zhang & Chen, 2018; Li et al., 2020), where an enclosing subgraph is extracted around each target link. Then the nodes in each enclosing subgraph are labeled differently according to their distances to the source and target nodes. Finally a GNN is applied to each enclosing subgraph to learn a link representation for link prediction. At first glance, both methods seem to learn graph structure features associated with the target link, and leverage these structure features for link prediction. However, as we will see, the two methods have **fundamentally different power** in terms of learning the *structural representations* of links.

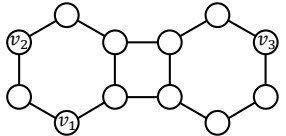

Figure 1: The structural roles of link $(v_1, v_2)$ and link $(v_1, v_3)$ are different, but GAE will assign equal probabilities to them.

We first show that by individually learning source and target node embeddings, GAE methods *cannot* differentiate links with different structural roles. To intuitively understand this, we give an example in Figure 1. In this graph, nodes $v_2$ and $v_3$ have the same structural roles (symmetric/isomorphic to each other). A GAE will learn the same node embeddings for $v_2$ and $v_3$, thus giving the same predicted probabilities for link $(v_1, v_2)$ and link $(v_1, v_3)$. However, the structural roles of link $(v_1, v_2)$ and link $(v_1, v_3)$ are apparently different – $v_1$ intuitively should have unequal probabilities connecting to $v_2$ and $v_3$. Next, we propose a *labeling trick*, which gives a label to each node as its additional feature, where the source and target nodes are labeled differently from the rest. We show that combined with the labeling trick, a sufficiently expressive GNN can learn the same representations for two links if and only if their structural roles are the same within the graph. This way, $(v_1, v_2)$ and $(v_1, v_3)$ will be predicted differently in Figure 1. We further show that SEAL is such an example. Finally, we give a more practical definition of isomorphism, called *local isomorphism*, which defines two nodes/links as isomorphic if their local neighborhood subgraphs are isomorphic. We argue that GNNs for link prediction should target on local-isomorphism-discriminating.

We conduct a thorough comparison among different link prediction methods, including SEAL and various GAE and embedding methods, on the recent large-scale Open Graph Benchmark (OGB) datasets (Hu et al., 2020). We show that SEAL with the labeling trick has up to 195% higher Hits@100 than GAE methods, achieving new state-of-the-art results on 3 out of 4 datasets.

## 2 PRELIMINARIES

In this section, we formally define the notions of graph, permutation, isomorphism, and GNN.

**Definition 1.** *(Graph). We consider an undirected graph $\mathcal{G} = (V, E, \mathbf{A})$, where $V = \{1, 2, \ldots, n\}$ is the set of $n$ vertices, $E \subseteq V \times V$ is the set of edges, and $\mathbf{A} \in \mathbb{R}^{n \times n \times k}$ contains the node and edge features with its diagonal components $\mathbf{A}_{i,i,:}$ denoting node attributes and off-diagonal components $\mathbf{A}_{i,j,:}$ denoting edge attributes. We further use $A \in \{0, 1\}^{n \times n}$ to denote the adjacency matrix of $\mathcal{G}$ with $A_{i,j} = 1$ iff $(i, j) \in E$. If there are no node/edge features, we let $\mathbf{A} = A$. Otherwise, $A$ can be regarded as the first slice of $\mathbf{A}$, i.e., $A = \mathbf{A}_{:,:,1}$.*

**Definition 2.** *(Permutation) A node permutation $\pi$ is a bijective mapping from $\{1, 2, \ldots, n\}$ to $\{1, 2, \ldots, n\}$. All $n!$ possible $\pi$'s constitute the permutation group $\Pi_n$. We define $\pi(S) = \{\pi(i) | i \in S\}$ when $S$ is a subset of $\{1, 2, \ldots, n\}$. We further define the permutation of $\mathbf{A}$ as $\pi(\mathbf{A})$, where $\pi(\mathbf{A})_{\pi(i), \pi(j),:} = \mathbf{A}_{i,j,:}$. In other words, $\pi(\mathbf{A})_{i,j,:} = \mathbf{A}_{\pi^{-1}(i), \pi^{-1}(j),:}$.*

**Definition 3.** *(Set isomorphism) Given two $n$-node graphs $\mathcal{G} = (V, E, \mathbf{A})$, $\mathcal{G}' = (V', E', \mathbf{A}')$, and two node sets $S \subseteq V$, $S' \subseteq V'$, we say $(S, \mathbf{A})$ and $(S', \mathbf{A}')$ are isomorphic (denoted by $(S, \mathbf{A}) \simeq (S', \mathbf{A}')$) if $\exists \pi \in \Pi_n$ such that $S = \pi(S')$ and $\mathbf{A} = \pi(\mathbf{A}')$.*

When $(V, \mathbf{A}) \simeq (V', \mathbf{A}')$, we say two graphs $\mathcal{G}$ and $\mathcal{G}'$ are *isomorphic* (abbreviated as $\mathbf{A} \simeq \mathbf{A}'$ because $V = \pi(V')$ for any $\pi$). Note that set isomorphism is **more strict** than graph isomorphism, because it not only requires graph isomorphism, but also requires the permutation maps a specific subset $S$ to another subset $S'$. When $S \subset V$ and $S' \subset V'$, we are often more concerned with the case of $\mathbf{A} = \mathbf{A}'$, where we are to find isomorphic node sets **in the same graph** (automorphism). For example, when $S = \{i\}, S' = \{j\}$ (single node case) and $(i, \mathbf{A}), (j, \mathbf{A})$ are isomorphic, it means $i$ and $j$ are on the same orbit of graph $\mathbf{A}$ (i.e., they have symmetric positions/same structural roles within the graph). An example is $v_2$ and $v_3$ in Figure 1.

**Definition 4.** *(Invariant function) A function $f$ defined over the space of $(S, \mathbf{A})$ is invariant if $\forall \pi \in \Pi_n, f(S, \mathbf{A}) = f(\pi(S), \pi(\mathbf{A}))$.*

**Definition 5.** *(GNN) A GNN is an invariant function mapping from the space of $(S, \mathbf{A})$ to $\mathbb{R}^d$. More specifically, a GNN first performs multiple invariant message passing operations to compute a node embedding $\mathbf{z}_i = \text{GNN}(i, \mathbf{A})$ for all $i \in S$, and then performs a set aggregation (pooling) over $\{\mathbf{z}_i | i \in S\}$, written as $\text{AGG}(\{\mathbf{z}_i | i \in S\})$, as the set $S$'s representation $\text{GNN}(S, \mathbf{A})$.*

Note that, when $|S| = 1$, the set aggregation is often an identity mapping. In graph classification ($S = V$), we use a graph pooling layer over node embeddings to compute the graph representation.

## 3 GAE AND STRUCTURAL LINK REPRESENTATION

In this section, we review how GAE methods predict links, and show that simply aggregating node embeddings learned by a GNN cannot lead to effective link representations. We use $\mathbf{A}$ to denote the incomplete network to perform link prediction.

### 3.1 GAE FOR LINK PREDICTION

Graph Autoencoder (GAE) methods (Kipf & Welling, 2016b) first use a GNN to compute a node embedding $\boldsymbol{z}_i$ for each node $i$, and then use $f(\boldsymbol{z}_i, \boldsymbol{z}_j)$ to predict the link $(i, j)$:

$$\hat{\boldsymbol{A}}_{i,j} = f(\boldsymbol{z}_i, \boldsymbol{z}_j), \text{ where } \boldsymbol{z}_i = \text{GNN}(i, \mathbf{A}), \ \boldsymbol{z}_j = \text{GNN}(j, \mathbf{A}) \tag{1}$$

where $\hat{\boldsymbol{A}}_{i,j}$ is the predicted score for link $(i, j)$. The model is trained to maximize the likelihood of reconstructing the true adjacency matrix. The original GAE uses a two-layer GCN (Kipf & Welling, 2016a) as the GNN, and let $f(\boldsymbol{z}_i, \boldsymbol{z}_j) := \sigma(\boldsymbol{z}_i^\top \boldsymbol{z}_j)$. In principle, we can replace GCN with any message passing neural network (Gilmer et al., 2017), and use an MLP over the aggregation of $\boldsymbol{z}_i$ and $\boldsymbol{z}_j$ as the $f(\boldsymbol{z}_i, \boldsymbol{z}_j)$. Popular aggregation functions include concatenation, mean and Hadamard product, etc. In the following, we will use GAE to denote a general class of GNN-based link prediction methods, without differentiating the specific choices of GNN and $f$.

### 3.2 GAE CAN LEARN STRUCTURAL NODE REPRESENTATIONS

Following (Srinivasan & Ribeiro, 2020; Li et al., 2020), we first define most expressive structural representations for nodes and links. Then we relate them to GAE-learned node embeddings and show that GAE is not capable of learning structural link representations.

**Definition 6.** *Given an invariant function* $\Gamma(\cdot)$, $\Gamma(S, \mathbf{A})$ *is a **most expressive structural representation** for* $(S, \mathbf{A})$ *if* $\forall (S, \mathbf{A}, S', \mathbf{A}')$, $\Gamma(S, \mathbf{A}) = \Gamma(S', \mathbf{A}') \Leftrightarrow (S, \mathbf{A}) \simeq (S', \mathbf{A}')$.

For simplicity, we will briefly use "structural representation" to denote most expressive structural representation in the rest of the paper. We will omit $\mathbf{A}$ if it is clear from context. We call $\Gamma(i, \mathbf{A})$ a *structural node representation* for $i$, and call $\Gamma(\{i, j\}, \mathbf{A})$ a *structural link representation* for $(i, j)$.

The above definition indicates that two node sets have the same structural representations if and only if they are isomorphic to each other. In the same graph $\mathbf{A}$, structural representations **uniquely** mark the structural roles of nodes or node sets. This is in contrast to positional node embeddings such as DeepWalk (Perozzi et al., 2014) and matrix factorization (Mnih & Salakhutdinov, 2008), where two isomorphic nodes can have different node embeddings (Ribeiro et al., 2017).

So why do we need to define structural representations? From a node classification point of view, it is because two isomorphic nodes in a network are perfectly symmetric to each other, and should be indistinguishable using any labeling functions on graphs (i.e., they should have the same ground truth $y$). Learning a structural node representation can guarantee that isomorphic nodes are always classified into the same class.

Then, a natural question to ask is, do GNNs learn structural node representations? The answer is no. Recall that $(i, \mathbf{A}) \simeq (j, \mathbf{A}') \Rightarrow \mathbf{A} \simeq \mathbf{A}'$. If a GNN can learn structural node representations, we can always use it for graph isomorphism test by checking whether there exist two nodes in two graphs sharing the same structural node representation. In fact, existing GNNs' graph discriminating power is bounded by the Weisfeiler-Lehman (WL) test (Morris et al., 2019; Maron et al., 2019), which provably fails to distinguish certain non-isomorphic graphs (Cai et al., 1992). Despite this, GNNs/WL are still **powerful enough** to learn representations that can **distinguish almost all non-isomorphic nodes and graphs** (Babai & Kucera, 1979).

For easy analysis, we assume there exists a *node-most-expressive* GNN that can output structural node representations thus able to distinguish all non-isomorphic nodes. Despite this, the techniques we will present are **not limited to** node-most-expressive GNNs, but also benefit practical GNNs.

**Definition 7.** *A GNN is **node-most-expressive** if* $\forall (i, \mathbf{A}, j, \mathbf{A}')$, $\text{GNN}(i, \mathbf{A}) = \text{GNN}(j, \mathbf{A}') \Leftrightarrow (i, \mathbf{A}) \simeq (j, \mathbf{A}')$.

Recall that GAE first uses a GNN to compute node embeddings. Therefore, GAE with a node-most-expressive GNN is able to leverage structural node representations for link prediction.

### 3.3 GAE CANNOT LEARN STRUCTURAL LINK REPRESENTATIONS

The next question to ask is whether GAE learns structural link representations. That is, does the aggregation of structural node representations of $i$ and $j$ result in a structural *link* representation of $(i, j)$? The answer is no, as shown in previous works (Srinivasan & Ribeiro, 2020; Zhang & Chen, 2020). We have also illustrated it in the introduction. In Figure 1, we have two isomorphic nodes $v_2$ and $v_3$, thus $v_2$ and $v_3$ will have the same structural node representation. By aggregating structural node representations as link representations, GAE will assign $(v_1, v_2)$ and $(v_1, v_3)$ the same link representation and predict them to have equal probabilities of forming a link. However, $(v_1, v_2)$ and $(v_1, v_3)$ apparently have different structural link representations, which indicates that

**Proposition 1.** *Even with a node-most-expressive GNN, GAE **cannot** learn structural link representations.*

The root cause of this problem is that GAE learns representations for the source and target nodes **individually**, without considering their relative positions and associations. For example, although $v_2$ and $v_3$ are perfectly symmetric in the graph, when considering the source node $v_1$ to predict link from, $v_2$ and $v_3$'s positions w.r.t. $v_1$ are no longer symmetric.

## 4 HOW TO LEARN STRUCTURAL LINK REPRESENTATIONS?

In this section, we discuss how to enable GNNs to learn structural link representations with a simple labeling trick, and show that SEAL is a valid example to learn structural link representations.

### 4.1 LABELING TRICK

We first introduce the *labeling trick*.

**Definition 8. (Labeling trick)** *Given $(S, \mathbf{A})$, we stack a diagonal labeling matrix $\boldsymbol{I}^{(S)} \in \mathbb{R}^{n \times n}$ in the third dimension of $\mathbf{A}$ to get a new $\tilde{\mathbf{A}} \in \mathbb{R}^{n \times n \times (k+1)}$, where $\boldsymbol{I}$ satisfies: $\forall \pi \in \Pi_n$, (1) $\boldsymbol{I}^{(S)} = \pi(\boldsymbol{I}^{(S')}) \Rightarrow S = \pi(S')$, and (2) $S = \pi(S'), \mathbf{A} = \pi(\mathbf{A}') \Rightarrow \boldsymbol{I}^{(S)} = \pi(\boldsymbol{I}^{(S')})$. A simplest labeling trick is to let $\boldsymbol{I}_{ii}^{(S)} = 1$ if $i \in S$ otherwise 0.*

Note that the notation $\tilde{\mathbf{A}}$ should actually depend on $S$. For convenience, we omit such dependence on $S$ and infer it from the context. The first condition in Definition 8 requires the labeling matrix to identify the target node set $S$; and the second condition requires the labeling trick to be permutation equivariant, i.e., when $(S, A)$ and $(S', A')$ are isomorphic under $\pi$, the corresponding nodes $i = \pi(j)$ always have the same label. Essentially, the labeling trick **distinguishes** nodes in $S$ from the rest nodes. It can be as simple as only giving label 1 to nodes in $S$ and otherwise 0. Next, we show that with the labeling trick, a node-most-expressive GNN can learn structural *link* representations.

**Theorem 1.** *If a GNN is node-most-expressive, then with an injective set aggregation function AGG in Definition 5, $\text{GNN}(S, \tilde{\mathbf{A}}) = \text{GNN}(S', \tilde{\mathbf{A}}') \Leftrightarrow (S, \mathbf{A}) \simeq (S', \mathbf{A}')$ for any $S, \mathbf{A}, S', \mathbf{A}'$.*

We include all the proofs in the appendix. Theorem 1 implies that $\text{GNN}(S, \tilde{\mathbf{A}})$ is a structural representation for $(S, \mathbf{A})$. Recall Definition 5 defines $\text{GNN}(S, \tilde{\mathbf{A}}) = \text{AGG}(\{\text{GNN}(i, \tilde{\mathbf{A}}) | i \in S\})$. The above theorem indicates that although directly aggregating structural node representations learned from the original graph $\mathbf{A}$ does not lead to structural link representations, an injective aggregation over the structural node representations learned from the *labeled graph* $\tilde{\mathbf{A}}$ does lead to structural link representations. How intuitively is it possible? Let's return to the example in Figure 1. When we want to predict link $(v_1, v_2)$, we can mark $v_1, v_2$ with a different label from the rest nodes, as shown by the different color in Figure 2 left. With the source and target nodes labeled, when the GNN is computing $v_2$'s embedding, it is also "aware" of the source node $v_1$, instead of the previous agnostic way that treats $v_1$ the same as other nodes. And when we want to predict link $(v_1, v_3)$, we can again mark $v_1, v_3$ with a different label, as shown in Figure 2 right. This way, $v_2$ and $v_3$'s structural node representations are no longer the same in the two differently labeled graphs because of their different relative positions w.r.t. $v_1$, and we are able to give different predictions to $(v_1, v_2)$ and $(v_1, v_3)$.

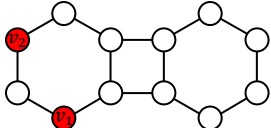 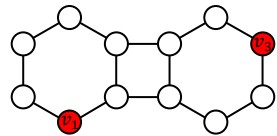

Figure 2: When we predict $(v_1, v_2)$, we will label these two nodes differently from the rest, so that a GNN is aware of the target link when computing $v_1$ and $v_2$'s embeddings. Similarly, when predicting $(v_1, v_3)$, nodes $v_1, v_3$ will be labeled differently. The aggregated embedding of $v_1, v_2$ in the left graph will be different from the aggregated embedding of $v_1, v_3$ in the right graph, enabling GNNs to predict $(v_1, v_2)$ and $(v_1, v_3)$ differently.

## 4.2 SEAL CAN LEARN STRUCTURAL LINK REPRESENTATIONS

In this section, we will review SEAL (Zhang & Chen, 2018; Li et al., 2020), and show that SEAL exactly uses a valid labeling trick that is able to learn structural link representations.

SEAL first extracts an *enclosing subgraph* ($h$-hop ego-network) around the link to predict.

**Definition 9.** *(Enclosing subgraph) Given* $(S, \mathbf{A})$*, the h-hop enclosing subgraph* $\mathbf{A}_S^{(h)}$ *of S is the subgraph induced from* $\mathbf{A}$ *by* $\cup_{i \in S} \{j | d(j, i) \leq h\}$*, where* $d(j, i)$ *is the shortest path distance between nodes* $j$ *and* $i$.

Then, SEAL applies a Double Radius Node Labeling (DRNL) to give an integer label to each node within the enclosing subgraph. DRNL assigns different labels to nodes with different distances w.r.t. both the source and target nodes, where the source and target nodes are always labeled 1, and nodes farther away from the source and target nodes get larger labels (starting from 2). For example, nodes with distances 1 and 1 to the source and target nodes will get label 2, and nodes with distances 1 and 2 to the source and target nodes will get label 3. So on and so forth. Finally the labeled enclosing subgraph is fed to a GNN to learn the link representation and output the probability of link existence.

It can be easily proved that DRNL satisfies the definition of the labeling trick, because node distances are invariant under permutation. DRNL not only differentiates the source and target nodes from the rest, but also differentiates nodes of different distances to the source and target nodes. The DRNL-labeled graphs also satisfy Theorem 1 and thus can enable a node-most-expressive GNN to learn structural link representations. Moreover, SEAL's distance-based node labeling scheme is formalized into *distance encoding*, or *DE*, in (Li et al., 2020), which theoretically shows that encoding distances between nodes can increase the representation power of normal GNNs.

SEAL only uses a subgraph $\mathbf{A}_S^{(h)}$ within $h$ hops from the source and target nodes instead of using the whole graph. This is for practical concerns (just like GAE typically uses less than 3 message passing layers in practice), and will be discussed in detail in Section 5. When $h \to \infty$, SEAL can also leverage the entire graph and learn structural representations for $(S, \mathbf{A})$:

**Proposition 2.** *When* $h \to \infty$*, SEAL can learn structural link representations with a node-most-expressive GNN.*

Note that although we show a node-most-expressive GNN combining with the labeling trick is able to learn structural link representations, **even without a node-most-expressive GNN, the labeling trick can still benefit most link representation learning tasks**. For example, in Figure 2, as long as a normal GNN can give different embeddings to $v_2$ and $v_3$ in the left and right graphs (which is easy for most GNNs), we can still differentiate link $(v_1, v_2)$ from link $(v_1, v_3)$. And this is not possible for GAE.

Despite the power, the labeling trick introduces extra computational complexity. The reason is that for every link $(i, j)$ to predict, we need to relabel the graph according to $(i, j)$. The same node $i$ will be labeled differently depending on the target link, and will be given a different node embedding by the GNN when it appears in different links' labeled graphs. This is different from GAE, where we do not relabel the graph and each node only has a single embedding vector. For a graph with $n$ nodes and $m$ links to predict, GAE needs to apply the GNN $\mathcal{O}(n)$ times to compute an embedding for each node, while SEAL needs to apply the GNN $\mathcal{O}(m)$ times for all links. When $m \gg n$, SEAL has worse time complexity than GAE, which is a trade-off for learning structural link representations.

## 5 LOCAL ISOMORPHISM: A MORE PRACTICAL VIEW OF ISOMORPHISM

In the previous analysis, we have assumed there exists a node-most-expressive GNN that can discriminate non-isomorphic nodes. Although a practical GNN can also discriminate almost all non-isomorphic nodes, there is little research discussing how to reach this discriminating power. In this section, we analyze GNNs for link prediction from a more practical point of view.

Practical GNNs usually simulate the 1-dimensional Weisfeiler-Lehman (1-WL) test (Weisfeiler & Lehman, 1968) to iteratively update each node's hidden state by aggregating its neighbors' states (we call them 1-WL-GNN). One most powerful 1-WL-GNN is the Graph Isomorphism Network (GIN) (Xu et al., 2018), which achieves theoretically the same discriminating power as 1-WL.

Although showing 1-WL-GNN's maximum discriminating power, previous GNN research has not discussed **how many message passing layers are required to reach this theoretical power**. We therefore introduce the following lemma.

**Lemma 1.** *Given a graph with $n$ nodes, a most powerful 1-WL-GNN takes up to $\mathcal{O}(n)$ message passing layers to discriminate all non-isomorphic nodes that 1-WL can discriminate.*

Lemma 1 suggests that 1-WL-GNN needs a number of message passing layers of the same order as the number of nodes in the graph to reach its maximum discriminating power. However, practical GNNs usually only use a small number of message passing layers (such as 1 or 2) while still achieving good performance. This perhaps suggests that in practice, we do not necessarily need to find nodes/links that are *exactly isomorphic* to each other. Instead, nodes/links which are *locally isomorphic* could even more strongly suggest they should be classified into the same class.

**Definition 10.** *(Local $h$-isomorphism)* $\forall(S, \mathbf{A}, S', \mathbf{A}')$, $(S, \mathbf{A})$ and $(S', \mathbf{A}')$ are locally $h$-isomorphic to each other if $(S, \mathbf{A}_S^{(h)}) \simeq (S', \mathbf{A}'^{(h)}_{S'})$.

For two sets $S, S'$, if their tuples with their $h$-hop enclosing subgraphs $(S, \mathbf{A}_S^{(h)})$ and $(S', \mathbf{A}'^{(h)}_{S'})$ are isomorphic, then we say they are *locally $h$-isomorphic*. We argue that this is a more useful definition than isomorphism, because isomorphism requires the entire graph looks identical from two sets' separate views, which is often too strict in practice. If we use the strict definition of isomorphism, we may fail to identify a lot of nodes/links that actually have very similar neighborhood structures. In fact, Babai & Kucera (1979) prove that at least $(n - \log n)$ nodes in almost all $n$-node graphs are *non-isomorphic* to each other. Therefore, it is less meaningful to only assign the same representations to nodes/links when they are strictly isomorphic, and GNNs targeting on isomorphism-discriminating tend to overfit. In comparison, local $h$-isomorphism only cares about whether the $h$-hop neighborhoods are isomorphic, which is more realistic and allows better generalizability.

With local $h$-isomorphism, all our previous conclusions using the standard isomorphism definition still apply. For example, GAE with a node-most-expressive GNN can be used to identify locally $h$-isomorphic nodes, but cannot discriminate locally $h$-non-isomorphic links. And a node-most-expressive GNN with the labeling trick (SEAL) can identify locally $h$-isomorphic node sets, etc. To enable a GNN to focus on local $h$-isomorphism-discriminating, all we need to do is to set a **boundary** between the $h$-hop enclosing subgraph and the rest of the graph, and apply the GNN only to the extracted subgraph within the boundary (possibly use more than $h$ message passing layers).

Finally, we want to answer a question: in what chance node local $h$-isomorphism also indicates link local $h$-isomorphism? If in a graph, two nodes being locally $h$-isomorphic almost always indicates their links with a third node are locally $h$-isomorphic, then learning structural link representations with SEAL may no longer be necessary, and using GAE to learn structural node representations may already be enough. In other words, we are interested in how often examples like Figure 1 appear in graphs. Fortunately, we have the following theorem, which shows that there are enough links where only identifying locally $h$-isomorphic nodes is not enough.

**Theorem 2.** *In any graphs with $n$ nodes and without node/edge features, if the degree of each node in the graph is between $1$ and $\mathcal{O}(\log^{\frac{1-\epsilon}{2h}} n)$ for any constant $\epsilon > 0$, then there exists $\omega(n^{2\epsilon})$ many pairs of nodes $u, v$ such that $u$ and $v$ are locally $h$-isomorphic while there exists another node $w$ such that $(u, w)$ and $(v, w)$ are not locally $h$-isomorphic.*

Table 1: Statistics and evaluation metrics of OGB link prediction datasets.

| Dataset | #Nodes | #Edges | Avg. node deg. | Density | Split ratio | Metric |
|---------|--------|--------|----------------|---------|-------------|--------|
| ogbl-ppa | 576,289 | 30,326,273 | 73.7 | 0.018% | 70/20/10 | Hits@100 |
| ogbl-collab | 235,868 | 1,285,465 | 8.2 | 0.0046% | 92/4/4 | Hits@50 |
| ogbl-ddi | 4,267 | 1,334,889 | 500.5 | 14.67% | 80/10/10 | Hits@20 |
| ogbl-citation | 2,927,963 | 30,561,187 | 20.7 | 0.00036% | 98/1/1 | MRR |

## 6 RELATED WORK

There are emerging interests in studying the expressive power of graph neural networks recently. Xu et al. (2018) and Morris et al. (2019) first show that the discriminating power of GNNs performing neighbor aggregation is bounded by the 1-WL test. Many works have since been proposed to increase the power of GNNs by simulating higher-order WL tests (Morris et al., 2019; Maron et al., 2019; Chen et al., 2019). However, most previous works focus on improving GNN's graph representation power. Little work has been done to analyze GNN's node/link representation power. Srinivasan & Ribeiro (2020) first formally studied the difference between structural representations of nodes and links. Although showing that structural node representations alone cannot perform link prediction, their way of learning structural link representations is to give up GNNs and instead use Monte Carlo samples of node embeddings learned by network embedding methods. In this paper, we show that *GNNs combined with a simple labeling trick can as well learn structural link representations*, which reassures using GNNs for link prediction.

Many works have implicitly assumed that if a model can learn node embeddings well, then combining the pairwise node embeddings can also lead to good link representations (Grover & Leskovec, 2016; Kipf & Welling, 2016b; Hamilton et al., 2017). However, we argue in this paper that by considering the source and target nodes explicitly with the labeling trick, we can aggregate better link representations from node embeddings. Li et al. (2020) studied one particular form of the labeling trick based on distance, and proved that distance encoding can improve 1-WL-GNNs' expressive power, enabling them to distinguish almost all $(S, \mathbf{A})$ tuples sampled from r-regular graphs. In this paper, we do not focus on improving 1-WL-GNNs' expressive power, but focus on how to enable a sufficiently expressive GNN to learn structural link representations in **any** graphs. We also provide a general definition of the labeling trick, and argue that a labeling trick as simple as only giving the source and target nodes 1 and other nodes 0 suffices to enable structural link representation learning.

## 7 EXPERIMENTS

In this section, we compare SEAL, GAE methods, and network embedding methods on the Open Graph Benchmark (OGB) (Hu et al., 2020) datasets. We use all the four link prediction datasets in OGB: `ogbl-ppa`, `ogbl-collab`, `ogbl-ddi`, and `ogbl-citation`. These datasets adopt realistic train/validation/test splitting methods, such as by resource cost in laboratory (`ogbl-ppa`), by time (`ogbl-collab` and `ogbl-citation`), and by drug target in the body (`ogbl-ddi`). They are also large-scale (up to 2.9M nodes and 30.6M edges), open-sourced, and have standard evaluation metrics, thus providing an ideal place to benchmark an algorithm's realistic link prediction power. The evaluation metrics include Hits@$K$ and MRR. Hits@$K$ counts the ratio of positive edges ranked at the K-th place or above against all the negative edges. MRR (Mean Reciprocal Rank) computes the reciprocal rank of the true target node against 1,000 negative candidates, averaged over all the true source nodes. Both metrics are higher the better. The statistics are in Table 1.

**Baselines.** We consider the following representative models. Except SEAL, all models first compute node embeddings from the original graph and use the Hadamard product between pairwise node embeddings as link representations. The link representations are fed to an MLP for final prediction.

- **MLP**: Input node features are directly used as the node embeddings.
- **Node2vec** (Perozzi et al., 2014; Grover & Leskovec, 2016): The node embeddings are the concatenation of node features and Node2vec embeddings.
- **MF**: Use free-parameter node embeddings trained end-to-end as the node embeddings.
- **GraphSAGE** (Hamilton et al., 2017): A GAE method with GraphSAGE as the GNN.

Table 2: Results for `ogbl-ppa`, `ogbl-collab`, `ogbl-ddi` and `ogbl-citation`.

| Method | ogbl-ppa Hits@100 (%) | | ogbl-collab Hits@50 (%) | | ogbl-ddi Hits@20 (%) | | ogbl-citation MRR (%) | |
|---|---|---|---|---|---|---|---|---|
| | Validation | **Test** | Validation | **Test** | Validation | **Test** | Validation | **Test** |
| **MLP** | 0.46±0.00 | 0.46±0.00 | 24.02±1.45 | 19.27±1.29 | – | – | 28.98±0.14 | 29.04±0.13 |
| **Node2vec** | 22.53±0.88 | 22.26±0.88 | 57.03±0.52 | 48.88±0.54 | 32.92±1.21 | 23.26±2.09 | 59.44±0.11 | 59.64±0.11 |
| **MF** | 32.28±4.28 | 32.29±0.94 | 48.96±0.29 | 38.86±0.29 | 33.70±2.64 | 13.68±4.75 | 53.11±5.65 | 53.16±5.65 |
| **GraphSAGE** | 17.24±2.64 | 16.55±2.40 | 56.88±0.77 | 48.10±0.81 | 62.62±0.37 | 53.90±4.74 | 82.17±0.86 | 82.28±0.84 |
| **GCN** | 18.45±1.40 | 18.67±1.32 | 52.63±1.15 | 44.75±1.07 | 55.50±2.08 | 37.07±5.07 | 84.49±1.08 | 84.56±1.10 |
| **GCN+LRGA** | 25.75±2.82 | 26.12±2.35 | 60.88±0.59 | 52.21±0.72 | **66.75**±0.58 | **62.30**±9.12 | 65.05±0.22 | 65.05±0.22 |
| **SEAL** | **51.25**±2.52 | **48.80**±3.16 | **63.89**±0.49 | **53.71**±0.47 | 28.49±2.69 | 30.56±3.86 | **85.09**±0.88 | **85.27**±0.91 |

- **GCN** (Kipf & Welling, 2016b): A GAE method with GCN as the GNN.
- **GCN+LRGA** (Puny et al., 2020): A GAE method with LRGA-module-enhanced GCN.
- **SEAL** (Zhang & Chen, 2018): Learn link representations from labeled subgraphs via a GNN.

All the GAE methods' GNNs have 3 message passing layers with 256 hidden dimensions, with a tuned dropout ratio in $\{0, 0.5\}$. The GNN in SEAL follows the original paper, which has 3 GCN layers with 32 hidden dimensions each, with a tuned subgraph hop $h$ in $\{1, 2\}$. The `ogbl-ddi` graph contains no node features, so MLP is omitted, and the GAE methods here use free-parameter node embeddings as the GNN input node features and trained them together with the GNN parameters. For SEAL, the DRNL node labels are input to an embedding layer and then concatenated with the node features (if any) as the GNN input. More details are in Appendix D.

**Results and discussion.** We compare the baselines' link prediction performance on `ogbl-ppa`, `ogbl-collab`, `ogbl-ddi`, and `ogbl-citation`. Among them, `ogbl-ppa` is a protein-protein association graph where the task is to predict biologically meaningful associations between proteins. `ogbl-collab` is an author collaboration graph, where the task is to predict future collaborations. `ogbl-ddi` is a drug-drug interaction network, where each edge represents an interaction between drugs which indicates the join effect of taking the two drugs together is considerably different from their independent effects. `ogbl-citation` is a paper citation network, where the task is to predict missing citations. We present the ten-time average results in Table 2. The results show that SEAL achieves the best performance on 3 out of 4 datasets. It outperforms GAE and network embedding methods, sometimes by surprisingly large margins. For example, in the challenging `ogbl-ppa` graph, SEAL achieves an Hits@100 of 48.80, which is over **50% higher** than the second-best baseline MF which only has an Hits@100 of 32.29. In comparison, all GAE methods achieve Hits@100 lower than 30. SEAL has **87%-195% improvement** over GAE methods in this dataset, which demonstrates the superiority of learning structural link representations over structural node representations for link prediction. SEAL also improves the state-of-the-art results for `ogbl-collab` and `ogbl-citation` in both validation and test performance.

Nevertheless, we observe that SEAL does not perform well on `ogbl-ddi`. `ogbl-ddi` is considerably denser than the other graphs. It only has 4,267 nodes, but has 1,334,889 edges, which results in an average node degree of 500.5 and density of 14.67%. Interestingly, although SEAL is able to beat MF and Node2vec, it falls behind GAE methods with free-parameter node embeddings. One possible reason is that the nodes in `ogbl-ddi` are so densely connected that a practical GNN with limited expressive power is hard to inductively learn any meaningful structural patterns. In comparison, the free-parameter node embeddings make GAE methods transductive and no longer focus on learning structural patterns, but focus on optimizing node embeddings. An interesting topic is thus how to improve structural representation learning on dense graphs, which we leave for future work.

## 8  CONCLUSIONS

In this paper, we have revisited the topic of using graph neural networks for link prediction. We reviewed two popular classes of methods, GAE and SEAL. We first showed that GAE methods cannot learn structural link representations by aggregating individually learned node embeddings. We further showed that combining with a simple labeling trick, a node-most-expressive GNN can learn structural link representations, and SEAL is such an example. Experiments on 4 large-scale OGB datasets demonstrate the superiority of learning structural link representations with SEAL.

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

## A    PROOF OF THEOREM 1

Recall Definition 5 defines:

$$\text{GNN}(S, \tilde{\mathbf{A}}) = \text{AGG}(\{\text{GNN}(i, \tilde{\mathbf{A}})|i \in S\}).$$

Thus, we only need to show $\text{AGG}(\{\text{GNN}(i, \tilde{\mathbf{A}})|i \in S\}) = \text{AGG}(\{\text{GNN}(i, \tilde{\mathbf{A}}')|i \in S'\})$ iff $(S, \mathbf{A}) \simeq (S', \mathbf{A}')$.

To prove the first direction, we notice that with an injective AGG,

$$\text{AGG}(\{\text{GNN}(i, \tilde{\mathbf{A}})|i \in S\}) = \text{AGG}(\{\text{GNN}(i, \tilde{\mathbf{A}}')|i \in S'\})$$

$$\implies \exists\, v_1 \in S, v_2 \in S', \text{ such that } \text{GNN}(v_1, \tilde{\mathbf{A}}) = \text{GNN}(v_2, \tilde{\mathbf{A}}') \tag{2}$$

$$\implies (v_1, \tilde{\mathbf{A}}) \simeq (v_2, \tilde{\mathbf{A}}') \quad \text{(because GNN is node-most-expressive)} \tag{3}$$

$$\implies \exists\, \pi \in \Pi_n, \text{ such that } v_1 = \pi(v_2), \tilde{\mathbf{A}} = \pi(\tilde{\mathbf{A}}'). \tag{4}$$

Remember $\tilde{\mathbf{A}}$ is constructed by stacking $\mathbf{A}$ and $\boldsymbol{I}^{(S)}$ in the third dimension, where $\boldsymbol{I}^{(S)}$ is a diagonal matrix satisfying: $\forall \pi \in \Pi_n$, (1) $\boldsymbol{I}^{(S)} = \pi(\boldsymbol{I}^{(S')}) \Rightarrow S = \pi(S')$, and (2) $S = \pi(S'), \mathbf{A} = \pi(\mathbf{A}') \Rightarrow \boldsymbol{I}^{(S)} = \pi(\boldsymbol{I}^{(S')})$. With $\tilde{\mathbf{A}} = \pi(\tilde{\mathbf{A}}')$, we have both

$$\mathbf{A} = \pi(\mathbf{A}'), \ \boldsymbol{I}^{(S)} = \pi(\boldsymbol{I}^{(S')}).$$

Because $\boldsymbol{I}^{(S)} = \pi(\boldsymbol{I}^{(S')}) \Rightarrow S = \pi(S')$, we have

$$\text{AGG}(\{\text{GNN}(i, \tilde{\mathbf{A}})|i \in S\}) = \text{AGG}(\{\text{GNN}(i, \tilde{\mathbf{A}}')|i \in S'\})$$

$$\implies \exists\, \pi \in \Pi_n, \text{ such that } S = \pi(S'), \mathbf{A} = \pi(\mathbf{A}') \tag{5}$$

$$\implies (S, \mathbf{A}) \simeq (S', \mathbf{A}'). \tag{6}$$

Now we prove the second direction. We have:

$$(S, \mathbf{A}) \simeq (S', \mathbf{A}')$$

$$\implies \exists\, \pi \in \Pi_n, \text{ such that } S = \pi(S'), \mathbf{A} = \pi(\mathbf{A}') \tag{7}$$

$$\implies \exists\, \pi \in \Pi_n, \text{ such that } S = \pi(S'), \boldsymbol{I}^{(S)} = \pi(\boldsymbol{I}^{(S')}), \mathbf{A} = \pi(\mathbf{A}') \tag{8}$$

$$\implies \exists\, \pi \in \Pi_n, \text{ such that } S = \pi(S'), \tilde{\mathbf{A}} = \pi(\tilde{\mathbf{A}}') \tag{9}$$

$$\implies \forall v_1 \in S, v_2 \in S', v_1 = \pi(v_2), \text{ we have } \text{GNN}(v_1, \tilde{\mathbf{A}}) = \text{GNN}(v_2, \tilde{\mathbf{A}}') \tag{10}$$

$$\implies \text{AGG}(\{\text{GNN}(v_1, \tilde{\mathbf{A}})|v_1 \in S\}) = \text{AGG}(\{\text{GNN}(v_2, \tilde{\mathbf{A}}')|v_2 \in S'\}), \tag{11}$$

which concludes the proof.

## B    PROOF OF LEMMA 1

We first note that for a most powerful 1-WL-GNN, after a message passing layer, gives different embeddings to any two nodes that 1-WL gives different colors to after one iteration. So we only need to show how many iterations 1-WL takes to converge in any graph.

Note that if two nodes are given different colors by 1-WL at some iteration (they are discriminated by 1-WL), their colors are always different in any future iteration. And if at some iteration, all nodes' colors are the same as their colors in the last iteration, then 1-WL will stop (1-WL fails to discriminate any more nodes and has converged). Therefore, before termination, 1-WL will increase its total number of colors by at least 1 after every iteration. Because there are at most $n$ different final colors given an $n$-node graph, 1-WL takes at most $n - 1 = \mathcal{O}(n)$ iterations before assigning all nodes different colors.

Now it suffices to show that there exists a $n$-node graph that 1-WL takes $\mathcal{O}(n)$ iterations to converge. Suppose there is a linked list of $n$ nodes (a path). Then by simple calculation, it takes $\lceil n/2 \rceil$ iterations for 1-WL to converge, which concludes the proof.

## C    Proof of Theorem 2

Our proof has two steps. First, we would like to show that there are $\omega(n^\epsilon)$ nodes that are locally $h$-isomorphic to each other. Then, we prove that among these nodes, there are at least $\omega(n^{2\epsilon})$ pairs of nodes such that there exists another node constructing locally $h$ non-isomorphic links with either of the two nodes in each node pair.

**Step 1.** Consider an arbitrary node $v$ and denote the subgraph induced by the nodes that are at most $h$-hop away from $v$ as $G_v^{(h)}$ (the $h$-hop enclosing subgraph of $v$). As each node is with degree $d = \mathcal{O}(\log^{\frac{1-\epsilon}{2h}} n)$, then the number of nodes in $G_v^{(h)}$, denoted by $|V(G_v^{(h)})|$, satisfies

$$|V(G_v^{(h)})| \leq \sum_{i=0}^{h} d^i = \mathcal{O}(d^h) = \mathcal{O}(\log^{\frac{1-\epsilon}{2}} n).$$

We set the max $K = \max_{v \in V} |V(G_v^{(h)})|$ and thus $K = \mathcal{O}(\log^{\frac{1-\epsilon}{2}} n)$.

Now we expand subgraphs $G_v^{(h)}$ to $\bar{G}_v^{(h)}$ by adding $K - |V(G_v^{(h)})|$ independent nodes for each node $v \in V$. Then, all $\bar{G}_v^{(h)}$ have the same number of nodes, which is $K$, though they may not be connected graphs.

Next, we consider the number of non-isomorphic graphs over $K$ nodes. Actually, the number of non-isomorphic graph structures over $K$ nodes is bounded by $2^{\binom{K}{2}} = \exp(\mathcal{O}(\log^{1-\epsilon} n)) = o(n^{1-\epsilon})$.

Therefore, due to the pigeonhole principle, there exist $n/o(n^{1-\epsilon}) = \omega(n^\epsilon)$ many nodes $v$ whose $\bar{G}_v^{(h)}$ are isomorphic to each other. Denote the set of these nodes as $V_{iso}$, which consist of nodes that are all locally $h$-isomorphic to each other. Next, we focus on looking for other nodes to form locally $h$-non-isomorphic links with nodes $V_{iso}$.

**Step 2.** Let us partition $V_{iso} = \cup_{i=1}^{q} V_i$ so that for all nodes in $V_i$, they share the same first-hop neighbor sets. Then, consider any pair of nodes $u, v$ such that $u, v$ are from different $V_i$'s. Then, we may pick one $u$'s first-hop neighbor $w$ that is not $v$'s first-hop neighbor. We know such $w$ exists because of the definition of $V_i$. As $w$ is $u$'s first-hop neighbor and is not $v$'s first-hop neighbor, $(u, w)$ and $(v, w)$ are not locally 1-isomorphic and thus not locally $h$-isomorphic. Therefore, based on this partition, we know the number of node pairs $u, v$ such that there exists another node $w$ making $(u, w)$ and $(v, w)$ not locally $h$-isomorphic is at least

$$Y \geq \prod_{i,j=1, i \neq j}^{q} |V_i||V_j| = \frac{1}{2}\left[(\sum_{i=1}^{q}|V_i|)^2 - \sum_{i=1}^{q}|V_i|^2\right]. \tag{12}$$

Because of the definitions of the partition, $\sum_{i=1}^{q}|V_i| = |V_{iso}| = \omega(n^\epsilon)$ and the size of each $V_i$ satisfies

$$1 \leq |V_i| \leq d_w = \mathcal{O}(\log^{\frac{1-\epsilon}{2h}} n),$$

where $w$ is one of the common first-hop neighbors shared by all nodes in $V_i$ and $d_w$ is its degree.

By plugging in the range of $|V_i|$, Eq.12 leads to

$$Y \geq \frac{1}{2}(\omega(n^{2\epsilon}) - \omega(n^\epsilon)\mathcal{O}(\log^{\frac{1-\epsilon}{2h}} n)) = \omega(n^{2\epsilon}),$$

which concludes the proof.

## D    More details about baselines

We include baselines achieving top performances on the OGB Link Prediction Leaderboard[1]. All the methods have their open-sourced code and paper available from the leaderboard. We adopt the numbers published on the leaderboard if available, otherwise we run the method ourselves using the

---

[1]https://ogb.stanford.edu/docs/leader_linkprop/

open-sourced code. For the baseline GCN+LRGA, its default hyperparameters result in out of GPU memory on `ogbl-citation`, even we use an NVIDIA V100 GPU with 32GB memory. Thus, we have to reduce its hidden dimension to 16 and matrix rank to 10. It is possible that it can achieve better performance with a larger hidden dimension and larger matrix rank using a GPU with a larger memory. Despite this, GCN+LRGA generally performs second to best among all methods.

We implemented SEAL using the PyTorch Geometric (Fey & Lenssen, 2019) package. Following the original paper (Zhang & Chen, 2018), we adopt a SortPooling layer (Zhang et al., 2018) after the GCN layers to readout the subgraph. For all datasets, SEAL only used a fixed 1% to 10% of all the available training edges as the positive training links, and sampled an equal number of negative training links randomly. SEAL showed excellent performance even without using the full training data, which indicates its strong inductive learning ability. Due to using different labeled subgraphs for different links, SEAL generally has longer running time than GAE methods. On the largest `ogbl-citation` graph, SEAL takes about 7 hours to finishing its training of 10 epochs, and takes another 28 hours to evaluate the validation and test MRR each. For `ogbl-ppa`, SEAL takes about 20 hours to train for 20 epochs and takes about 4 hours for evaluation. The other two datasets are finished within hours. We will release all our code for reproducing the experimental results.

