# OpenReview forum: "Revisiting Graph Neural Networks for Link Prediction"
_ICLR.cc/2021/Conference — Reject_

### Official Review · AnonReviewer1 · 2020-10-21
**Interesting summary of the field, but not enough contribution for the conference**

**Rating:** 3
**Confidence:** 4

**Review:**

Summary:

The paper presents results of two popular classes of methods for graph neural networks.  Namely, GAE and SEAL.  The paper show results on the Open Graph Benchmark of several existing methods, and concludes that GAE cannot learn structural link representations while SEAL can.

The paper is descriptive in its presentation, and does a good work on detailing the ideas behind GAE and SEAL.  However, I felt the lack of a contribution on the paper.  As it stands, it reads more as a tutorial and presents key insights of existing methods.

Pros:
- Clear explanations of existing work.

Cons:
- No clear contribution over explaining existing approaches.
- Experiments show existing results on OGB.
- No results on the mentioned labeling trick are given.

Comments:
- In Fig. 1 you mentioned that $v_2$ and $v_3$ will get the same representation through a GAE.  Did you train one and observed it?  It will be more conclusive if you show the results and the embeddings to validate your observation.

- Your labeling trick depends on the set of nodes $S$ used.  However, how do you select such sets to produce the labeling? Are you sampling all possible subsets $S$?  Since this seems to be a form of contribution, it should be clear how to use it in any scenario.

- In your experiments, due to the way the paper presents the label trick as the novelty, I was expecting to see the boost of performance of such trick on the existing architectures.  However, Section 7 presents results of existing approaches on the OGB.

Minor comments:
- "Following (Srinivasan & Ribeiro, 2020; Li et al., 2020)" should be a textual citation

Overall rating:
Due to the main problems stated, I cannot recommend the paper for publication at ICLR.  The manuscript reads more like a tutorial, which may be of interest for readers finding a summary of the advances.  However, as a paper advancing the field of graph neural networks I feel it lacking.

---

> ### Author Response · Authors · 2020-11-15
> **The paper is not just a tutorial of existing methods.**
>
> We thank the reviewer for the comments. However, we do not agree that this paper is only a tutorial of existing methods. Our paper discusses the deep differences between GAE and SEAL, and answers why SEAL outperforms GAE on link prediction using extensive theoretical arguments. Also, it seems the reviewer has missed a lot of important details in the paper. Below we try to address the comments.
>
> 1."No clear contribution over explaining existing approaches."
>
> We argue that the main contribution of the paper is to demonstrate why SEAL is able to outperform GAE methods on link prediction, and reveal that the key component is the labeling trick. It defines a general form of the labeling trick (Definition 5), and shows that with the labeling trick a sufficiently expressive GNN can discriminate all non-isomorphic links (Theorem 1), which is not possible for GAEs as shown in Figure 1. We also demonstrate why practical GNNs only need a small local subgraph to achieve a good performance without reaching the theoretical worst bound of WL test through Definition 10 and Theorem 2. It is unfortunate that the reviewer overlooked these significant theoretical contributions.
>
> 2."Experiments show existing results on OGB."
>
> The results of SEAL on OGB leaderboard are uploaded by us after the deadline of ICLR 2021, and are used to support this submission. In this paper, we reimplemented SEAL, and fundamentally improved its scalability in order to run on OGB datasets. We are sorry to cause the confusion that it seems we are “copying” existing results on OGB leaderboard without doing experiments in this paper.
>
> 3."No results on the mentioned labeling trick are given."
>
> The entire section 4.2 is trying to convey that SEAL is exactly a GNN enhanced by labeling trick. Our results on SEAL show that a GCN enhanced by labeling trick outperforms baselines significantly.
>
> 4."In Fig. 1 you mentioned that v2 and v3 will get the same representation through a GAE. Did you train one and observed it? It will be more conclusive if you show the results and the embeddings to validate your observation."
>
> No, we didn’t. And it is not necessary. The reason is as follows. GNN has the same convolution parameters for all nodes. v2 and v3 also have the same neighborhood structure. Applying GNN on v2 and v3 will lead to exactly the same representations. The training of GNN will change the parameters over epochs, but still the parameters are shared across all nodes. So no matter how we train the GNN, v2 and v3 will always have the same representation.
>
> 5."Your labeling trick depends on the set of nodes S used. However, how do you select such sets to produce the labeling? Are you sampling all possible subsets S? Since this seems to be a form of contribution, it should be clear how to use it in any scenario."
>
> S is the set of nodes to learn a joint structural representation for. For link prediction, S is exactly the source and target nodes to predict link between. The notation of S is used consistently across the paper to denote the target node set of interest since Definition 3. There is no sampling needed.
>
> 6."In your experiments, due to the way the paper presents the label trick as the novelty, I was expecting to see the boost of performance of such trick on the existing architectures. However, Section 7 presents results of existing approaches on the OGB."
>
> SEAL is exactly a GCN + labeling trick. By showing SEAL’s significant performance boost over baselines, the experiments demonstrate the of great value of the labeling trick.
>
> 7."Following (Srinivasan & Ribeiro, 2020; Li et al., 2020)" should be a textual citation
>
> Thanks! We will revise it.
>
> Summarization
>
> We are again sorry about causing those confusions to the reviewer. To clarify better, SEAL is originally proposed in (Zhang and Chen NeurIPS 2018). It is a SotA method for link prediction, but people have little understanding of why it outperforms normal GNNs. In this paper, we identify one key component of SEAL is its labeling trick, and analyze theoretically how the labeling trick helps a GNN (that only discriminates non-isomorphic nodes) to also discriminate non-isomorphic links. We also reimplemented SEAL using advanced data structures and libraries to achieve ~1000 times speed up, and for the first time got its performance on the large-scale OGB datasets, which again verified its SotA performance. All these works are nontrivial, and beyond just a tutorial of existing methods.

---

> > ### Comment · AnonReviewer1 · 2020-11-17
> > **Some of my questions were addressed.  But I'm still concerned about the experimental contribution of the paper**
> >
> > 1. On the contribution.
> >
> > If you claim that the labeling trick is the main proposal of the paper, then it should take a leading role in the exposition of the manuscript and the results.  Moreover, this claim is limited since it was originally proposed by Zhang and Chen, as you mentioned.  Thus, showing that it works in general could be a contribution, but I don't see that in the current experiments.
> >
> > I cannot agree with the claim that the paper demonstrates the effectiveness of the trick wrt different architectures.  Are the experiments on Table 2 trying to show the different architectures with the labeling trick?  Because, I don't get that impression from reading that section.  I see from your response to my question 6, that SEAL is the only one doing what I'm asking, why not expand the test to the other architectures?  Did I misinterpret those experiments?
> >
> > An interesting ablation study could be performed in which you select the different architectures with and without the labellings and show their contribution.  Moreover, different amounts of intersection in the labellings could provide some insight on the performance of the networks in the particular structures that were intersected in the labels and, I would guess, a lower performance on their prediction.
> >
> > 2. Experiments on OGB.
> >
> > You should be clearer in the description of the paper.  Also, these results relate to my question above showing the contribution of the labeling trick and the methods.  Since you re-implemented them, are you doing what I asked before?
> >
> > 3. Experiments on the labeling trick
> >
> > As you see in my previous responses, what I was asking was experimental results showing the labeling trick at work.  For instance, you could take the architecture with and without the labeling trick on the OGB dataset and show the results on them.  These type of experiments are more convincing to show the effectiveness of your proposal on a set of architectures and data setups.
> >
> > 4. Similar representations on isomorphic nodes
> >
> > Similar to the discussion from [Guillaume Salha](https://openreview.net/forum?id=8q_ca26L1fz&noteId=a8XZXV1wIil), I see that there is a difference between the representation learned by the GAE.  You should be clearer on your description when talking about the node embedding and the structural representation.  I still, however, do not see how the structural representation can be exactly the same.  Could you do an experiment similar to what Guillaume did and show it for your arguments?
> >
> > Moreover, I did not follow your comments on the one-hot representations.   Could you please elaborate?
> >
> > I think that if you are making some assumptions on the type of encodings and representations, that should be clearly stated on the manuscript.
> >
> > 5. Consider adding such explanation on the paper.  It is not totally clear from your current description.
> >
> > 6. Thanks for the clarification.  See my questions regarding the experiments on the previous questions.
> >
> > Summary
> >
> > I thank the authors for the reply.  It helped me improve my understanding of the paper.  I still have some doubts on the experimental setup that demonstrates that the labeling trick is boosting the performance.  An ablation on different architectures with the trick would help to solidify this claim.  Moreover, the claim on the labeling trick is limited since it was previously proposed by Zhang and Chen.

---

> > > ### Author Response · Authors · 2020-11-18
> > > **Thank you for your quick response [1/2]**
> > >
> > > Thank you for your quick response! We answer your questions below.
> > >
> > > 1.Labeling trick was originally proposed by Zhang and Chen, thus is of limited contribution
> > >
> > > The labeling trick is NOT proposed by (Zhang and Chen NeurIPS 2018). (Zhang and Chen NeurIPS 2018) proposed SEAL which only used one particular form of the labeling trick, DRNL. And later (Li et. al. NeurIPS 2020) extended it to distance encoding. The exact labeling trick is proposed in this paper, which unifies the various node labeling schemes to a most general form (Definition 8), and proves that as long as a node labeling satisfies Definition 8, it can enable a sufficiently expressive GNN to discriminate all non-isomorphic links. Thus, identifying the most general form of the labeling trick and proves its great value for link prediction is our main contribution.
> > >
> > > 2."I cannot agree with the claim that the paper demonstrates the effectiveness of the trick wrt different architectures. Are the experiments on Table 2 trying to show the different architectures with the labeling trick?"
> > >
> > > We demonstrate the effectiveness of the trick for general message passing GNNs (Definition 5) theoretically. As long as a GNN satisfies Definition 5 and is node-most-expressive, the labeling trick enables it to discriminate all non-isomorphic links, which is not possible before applying the labeling trick as shown in Figure 1. In Table 2, we only show results of SEAL, because SEAL is GCN + labeling trick, which can represent a broad class of message passing based GNNs. Compared to pure GCN, SEAL shows up to ~200% performance gains.
> > >
> > > But we do agree that it's worth showing more architectures' performances with the labeling trick. Thus, we redo the experiments on ogbl-collab with GraphSAGE + labeling trick. It achieved 62.65 ± 0.23 validation Hits@50 and 52.11 ± 0.24 test Hits@50, which also outperformed pure GraphSAGE (56.88 ± 0.77, 48.10 ± 0.81) significantly. We are running more experiments. But we found labeling trick is generally helpful no matter what underlying GNN is used. We will add more ablation studies.
> > >
> > > 3."You should be clearer on your description when talking about the node embedding and the structural representation. I still, however, do not see how the structural representation can be exactly the same. Could you do an experiment similar to what Guillaume did and show it for your arguments?"
> > >
> > > We indeed discussed the difference between structural representation and positional node embeddings in section 3.2. We emphasized that structural representations uniquely mark the structural roles of nodes, where isomorphic nodes always have the same structural representation. On the contrary, positional node embeddings (such as MF, node2vec) do not hold this property -- two isomorphic nodes can have different positional embeddings.
> > >
> > > Following your suggestion, we did an experiment similar to what Guillaume did without using one-hot node features. The learned node representations for v2 and v3 are as follows:
> > >
> > > Epoch 1:
> > >
> > > v2: [-0.0317,  0.0142, -0.0031,  ...,  0.0187,  0.0027, -0.0844],
> > >
> > > v3: [-0.0317,  0.0142, -0.0031,  ...,  0.0187,  0.0027, -0.0844],
> > >
> > > Epoch 2:
> > >
> > > v2: [-0.0137,  0.0216,  0.0101,  ...,  0.0222, -0.0108, -0.0668],
> > >
> > > v3: [-0.0137,  0.0216,  0.0101,  ...,  0.0222, -0.0108, -0.0668],
> > >
> > > ...
> > >
> > > Epoch 10:
> > >
> > > v2: [ 0.0556,  0.0177,  0.1064,  ...,  0.0725, -0.1680, -0.0197],
> > >
> > > v3: [ 0.0556,  0.0177,  0.1064,  ...,  0.0725, -0.1680, -0.0197].
> > >
> > > At every epoch, v2 and v3 have exactly the same node representations, which verified that GAE always learns the same embeddings for isomorphic nodes.

---

> > > > ### Author Response · Authors · 2020-11-18
> > > > **Thank you for your quick response [2/2]**
> > > >
> > > > 4."Moreover, I did not follow your comments on the one-hot representations. Could you please elaborate?"
> > > >
> > > > We said that "it is meaningless to study the GNN representation power with one-hot node features". This is because, GNN representation power is measured by its ability to discriminate non-isomorphic nodes or non-isomorphic graphs from graph structures and node features. For example, in Figure 1, v1 and v2 are non-isomorphic, while v2 and v3 are isomorphic. If you run a GNN for enough iterations, it will learn different embeddings for v1 and v2, and learn the same embedding for v2 and v3. However, if you use one-hot features, you are assuming all nodes have different features initially, thus all nodes become non-isomorphic to each other immediately. Then, there is no need to run GNN anymore, because the initial one-hot features already discriminate all nodes. But why do we still study GNNs given we can use one-hot features? That is because although one-hot feature discriminates all nodes, it also loses GNN's ability to map isomorphic nodes (v2, v3) to the same embedding. In many cases we want this property because isomorphic nodes are perfectly symmetric in a graph -- they are structurally equivalent.
> > > >
> > > > Summary
> > > >
> > > > We thank R1 for rereading the paper. We will make the assumptions more clear and add more ablation studies. We respectfully argue our main contribution is the new theorems characterizing the great value of labeling trick to link prediction. Our experiment tried to preserve the original setting of SEAL as much as possible to respect previous work. It achieved three 1st places in the OGB leaderboard, which itself is a great contribution to the field and is among the very few papers that really advanced the OGB results in all ICLR submissions.

---

> > > > > ### Comment · AnonReviewer1 · 2020-11-20
> > > > > **Thanks for the clarification, but I still find the contribution not strong enough**
> > > > >
> > > > > I thank the authors for the clarifications.  I understand that your labeling trick is generalizing thus introduced by Zhang and Chen and Li et al.  However, I still find the contribution too weak and the experiments confusing.  If you are generalizing these labeling setups, it will be better to show how these instances can be applied and how much they enhance the architectures they are applied on.  Moreover, since you are generalizing the previous insights, it will be interesting to propose some labeling schemes and compare them with the existing ones for further insights within the labeling trick.
> > > > >
> > > > > I suggest you to perform a full ablation with the **different architectures** and **proposals for your labeling trick** to show it's proper enhancement capabilities.  I understand and appreciate the effort of running the experiments during this rebuttal period.  However, I don't encourage you to rush them.  Instead, define your experimental setup with calm and perform the experiments to improve your theoretical contribution.
> > > > >
> > > > > Regarding the embeddings you showed, does the predicted links differentiate between them?  Adding these results to your discussion could improve the paper.

---

### Official Review · AnonReviewer2 · 2020-10-21
**Need more experiment and comparison to support the argument**

**Rating:** 5
**Confidence:** 4

**Review:**

The paper focuses on the link prediction task for graph neural networks. More specifically, it compares GAE and SEAL by providing theoretical evidence why GAE is not able to learn structural link representations, which as a result leads to suboptimal performance in the link prediction task. The paper also introduces a labeling trick that can help GNNs to learn structural link representations.

Strong points:
1. The paper is clear, easy to understand, and the definitions and theorems provide good reference and proofs.
2. A deeper understanding of the link prediction task is indeed needed as a lot of GNN works are mostly using classification as the downstream task.
3. It provides good argument and clear example why some GNN models would not work well in link prediction by showing that individually learning node representations cannot handle the case where two nodes appear in topologically identical neighborhoods in the graph. It also provides insights as to how to mitigate such issues.

Weak points:
1. The comparison is oversimplified and overly generalized. The paper mostly compares GAE and SEAL, but there are much more GNN models that should be compared.
2. The sheer separation of GAE and SEAL model is not convincing. GAE and SEAL are both GNN models which follow the message passing and aggregation approaches. There's no clear argument or theoretical explanation how these two are fundamentally different.
3. There are a lot of existing work that considers position in encoding nodes (such as position aware gnn [2]) and should be able to learn expressive structural link representations. The paper does not compare with them.
4. The experiment does not cover the most recent models. In addition, more benchmark datasets should be used. For example, FB15K (this is a widely used knowledge graph link prediction benchmark dataset), PPI, WN18, etc. Since previous work on gnn link prediction uses
those dataset, I'm expecting to see a more comprehensive comparison using those datasets. More dataset can be found here: https://paperswithcode.com/task/link-prediction
5. The paper should also consider comparing with knowledge graph embedding models which are focusing on the link prediction task. There are also existing work that combine the idea of knowledge graph embedding with GNN models in link prediction [1].

Other comments:
1. In def 5, gnn does not always have an invariant function mapping, depending on the message passing operations used e.g. in the graph sage paper, there are comparison using lstm message passing operator, which is not permutation invariant.
2. Section 3.1 mentions that the paper uses GAE to denote the general class of GNN-based link prediction regardless of the choice of GNN and aggregation function. However, research has shown that the specific choices of message passing operations and aggregations have significant impact on the expressiveness of GNN [3]
3. Section 3.3 mentions that GAE cannot learn structural link representations, but from the example in fig1, it seems like this is caused by the graph isomorphism issue. there are existing work that can mitigate this issue, for example position-aware graph neural network [2].


Reference:
1. Schlichtkrull, Michael, et al. "Modeling relational data with graph convolutional networks." European Semantic Web Conference. Springer, Cham, 2018.
2. You, Jiaxuan, Rex Ying, and Jure Leskovec. "Position-aware graph neural networks." arXiv preprint arXiv:1906.04817 (2019).
3. Xu, Keyulu, et al. "How powerful are graph neural networks?." arXiv preprint arXiv:1810.00826 (2018).

---

> ### Author Response · Authors · 2020-11-15
> **Thank you.**
>
> We thank the reviewer for the comments.
>
> 1."The paper mostly compares GAE and SEAL, but there are much more GNN models that should be compared."
>
> GAE and SEAL are two main GNN link prediction paradigms. They do not restrict which specific GNN model is used. Any GNN, such as GCN, GraphSage or GIN, can be used in both GAE and SEAL frameworks. In the experiments, GCN, GraphSAGE and GCN+LRGA are used in GAE, and GCN is used in SEAL. Using more recent GNNs may further improve both GAE and SEAL's performance, but it is beyond the purpose of the paper (which compares the frameworks of GAE and SEAL).
>
> 2."The sheer separation of GAE and SEAL model is not convincing. GAE and SEAL are both GNN models which follow the message passing and aggregation approaches. There's no clear argument or theoretical explanation how these two are fundamentally different."
>
> We believe the entire section 3 and 4 are trying to discuss the differences between GAE and SEAL and answer why SEAL is more suitable than GAE for link prediction in a theoretical way. In short, SEAL uses GNN + labeling trick, while GAE only uses GNN.
>
> 3."There are a lot of existing work that considers position in encoding nodes (such as position aware gnn [2]) and should be able to learn expressive structural link representations. The paper does not compare with them."
>
> Thanks. We indeed think P-GNN can be a competitive baseline. However, the original P-GNN paper uses datasets with at most 3,000 nodes. It is nontrivial to scale it to OGB datasets with millions of nodes and tens of millions of edges, especially when one need to compute the shortest path distance and communicate between a node and all anchor nodes in a large graph. So are other possible baselines. Therefore, we only compared with the baseline results reported in the [OGB leaderboard](https://ogb.stanford.edu/docs/leader_linkprop/). We believe the purpose of such leaderboards is to provide a unified protocol/place for fairly comparing various GNNs, so that the authors can focus on improving their own model instead of spending great time addressing baselines’ different settings. Nevertheless, we will try to make P-GNN work on OGB and compare it in the revision.
>
> 4."Why don’t compare with knowledge graph embedding methods and use knowledge graph datasets?"
>
> Thanks. Knowledge graphs (FB15K, WN18 etc.) are heterogeneous, featureless, and each node has a distinct identity (type). These properties make knowledge graph link prediction very different from the standard link prediction problem in homogeneous networks. The baselines for knowledge graph link prediction and standard link prediction are not interchangeable. We can neither compare GAE/SEAL using knowledge graph datasets, nor compare with knowledge graph link prediction baselines using OGB.
>
> 5."In def 5, gnn does not always have an invariant function mapping, depending on the message passing operations used e.g. in the graph sage paper, there are comparison using lstm message passing operator, which is not permutation invariant."
>
> Thanks. Our definition of GNN covers a class of most popular permutation invariant message passing networks. The definition does not cover GraphSAGE with LSTM operator. However, such non-permutation-invariant versions are of less practical usefulness as two isomorphic structures can be mapped to different representations.
>
> 6."Section 3.1 mentions that the paper uses GAE to denote the general class of GNN-based link prediction regardless of the choice of GNN and aggregation function. However, research has shown that the specific choices of message passing operations and aggregations have significant impact on the expressiveness of GNN [3]"
>
> We agree with this. However, our Definition 5 covers the most expressive GNN, GIN, discussed in [3]. As long as the GNN used in GAE satisfies Definition 5, it is not able to discriminate between link (v1, v2) and link (v1, v3) in Figure 1 (even using GIN), while the same GNN + labeling trick can. In other words, regardless the expressiveness of the underlying GNN, our labeling trick can always improve its link prediction ability.
>
> 7."Section 3.3 mentions that GAE cannot learn structural link representations, but from the example in fig1, it seems like this is caused by the graph isomorphism issue. there are existing work that can mitigate this issue, for example position-aware graph neural network [2]."
>
> Thanks. We agree P-GNN is another way to break the symmetry in Figure 1. However, P-GNN introduces positional embeddings to GNN, making the GNN model no longer inductive. It also cannot guarantee to map two isomorphic links (with different relative positions to the anchor nodes) to the same representation, which is what labeling trick means to ressolve.
>
> [2] You, Jiaxuan, Rex Ying, and Jure Leskovec. "Position-aware graph neural networks." arXiv preprint arXiv:1906.04817 (2019).
> [3] Xu, Keyulu, et al. "How powerful are graph neural networks?." arXiv preprint arXiv:1810.00826 (2018).

---

### Official Review · AnonReviewer4 · 2020-10-24
**Review #4**

**Rating:** 4
**Confidence:** 3

**Review:**

This paper provides theoretical analysis of graph neural networks for link prediction, following a number of recent papers that have developed the theoretical understanding of graph neural networks. For example, the work of Xu et al (ICLR 2019) draws on the Weisfeiler-Lehman graph isomorphism test to develop a theoretical framework in which to analyse the expressive power of Graph Neural Networks. In particular, they show that architectural features in common graph neural network models limit the expressivity with respect to problems of node and graph classification and show how one can construct graph neural networks whose expressivity matches the Weisfeiler-Lehmam test. In more recent work, Li et al (NeurIPS 2020) show that augmenting node features with a "distance encoding" enables a graph neural network to distinguish node sets in cases where the Weisfeiler-Lehman test fails. This submission is perhaps most closely related to this recent paper of Li et al.

In this work, the authors discuss how GAEs cannot always discriminate between links (node pairs), even when a GAE has maximal expressivity at the node level (it can generate representations which discriminate all non-isomorphic nodes). They provide an enhancement they call a "labelling trick" which can be used to enable GAEs to discriminate between node sets. With the labelleing trick, ach node in the GAE has an additional feature corresponding to a node set specific label. That is, when learning representations for a given node set S, this feature takes on a value which is dependent on the node set S and the graph structure. The authors show (Theorem 1) that this enhancement enables any maximally node expressive GNN (a GNN which can discriminate all non-isomorphic nodes) to discriminate between any pair of non-isomorphic node sets. The strong performance of SEAL, a recent SOTA method for link prediction, can be understood in the context of this result.

As a second theoretical contribution, the authors define a "local-h isomorphism" concept, which they propose can be used to explain why practical GNN implementations perform well, despite theoretically requiring the number of layers to be proportional to the number of nodes in order to distinguish all nodes that 1-WL can discriminate (in the worst case).

**Positives**
Studying the expressive power of Graph Neural Network architectures is an important topic, as can be seen by the increasing number of papers in machine learning conferences over the past couple of years. The analysis in this paper extends the theory with respect to the link prediction problem and provides theoretical justification for the strong performance of SEAL on link prediction benchmarks.

**Concerns**
However, I have some concerns regarding the novelty of the contribution and the significance of the main theoretical result (Theorem 1).
1. The main methodological contribution in this paper is the concept of the labelling trick, which can be used to improve the representational power of GNNs and explain the strong performance of SEAL. This looks like a special case of the distance encoding method presented in Li et al (NeurIPS 2020). The authors cite this work but do not discuss how their labelling trick relates to the distance encoding. Li et al also provide an explanation of the strong performance of SEAL (as an instance of their distance encoding method) as well as using their analysis to motivate alternative models which give a small performance improvement over SEAL.
2. The main theoretical result (Theorem 1) assumes the existence of a GNN with maximum node expressivity, in the sense that it can discriminate all non-isomorphic nodes in a graph. By my understanding, this architecture does not exist. Indeed, I believe Theorem 3.7 of Li et al (NeurIPS 2020) shows that the distance encoding method (which seems to be closely related to the labelling trick) is not sufficient for this. In my view, a more useful theoretical result would be something along the lines of Theorem 3.3 of Li et al, which illustrates how the distance encoding method improves the representational power of GNNs for a large number of graphs where methods bounded by 1-WL would fail.

---

> ### Author Response · Authors · 2020-11-15
> **Distance encoding is a special labeling trick. Theorem 1 is also useful for practical GNNs.**
>
> We thank the reviewer for the insightful comments. Below we address the main concerns.
>
> 1."The main methodological contribution in this paper is the concept of the labelling trick, which can be used to improve the representational power of GNNs and explain the strong performance of SEAL. This looks like a special case of the distance encoding method presented in Li et al (NeurIPS 2020). The authors cite this work but do not discuss how their labelling trick relates to the distance encoding."
>
> We indeed discussed the relation between our paper and (Li et. al. NeurIPS 2020) in the Related Work section. To restate, distance encoding is a particular form of the labeling trick. When distance encoding is used for link prediction, the shortest-path-distance-based node labels are equivalent to the DRNL of SEAL (both label a node according to its SPD to the source and target nodes). The difference between our paper and (Li et. al. NeurIPS 2020) is that: (Li et. al. 2020) shows that distance encoding can improve 1-WL-GNN’s expressive power, enabling them to distinguish almost all node sets sampled from *r-regular* graphs. However, our analysis is not restricted to r-regular graphs, but discusses any general graph. We show that a sufficiently expressive GNN can discriminate non-isomorphic node sets in *any graphs* with the labeling trick. Thus, our paper gives a broader applicability of the labeling trick, showing that it is generally useful for link prediction in any graphs.
>
> Moreover, we give a more general definition of labeling trick, which incorporates DRNL, distance encoding, and the simplest annotation of S, etc. into the same framework. As long as a node labeling satisfies Definition 8, it can enable the structural link representation learning ability of GNNs. Thus, it actually does not restrict the node labels to be distance-based, and implies a promising research field of studying more novel ways of node labeling to enhance GNNs.
>
> 2."The main theoretical result (Theorem 1) assumes the existence of a GNN with maximum node expressivity, in the sense that it can discriminate all non-isomorphic nodes in a graph. By my understanding, this architecture does not exist. Indeed, I believe Theorem 3.7 of Li et al (NeurIPS 2020) shows that the distance encoding method (which seems to be closely related to the labelling trick) is not sufficient for this. In my view, a more useful theoretical result would be something along the lines of Theorem 3.3 of Li et al, which illustrates how the distance encoding method improves the representational power of GNNs for a large number of graphs where methods bounded by 1-WL would fail."
>
> We actually discussed this issue in the paragraph after Proposition 2. We emphasized that even without a node-most-expressive GNN, the labeling trick can still benefit most link representation learning tasks. For example, in Figure 2, as long as a normal GNN can give different embeddings to v2 and v3 in the left and right graphs (which is easy for most GNNs), we can still differentiate link (v1, v2) from link (v1, v3) because the neighborhood of v2 and v3 are no longer the same with the node label of v1.
>
> Also, as discussed in the paper, although such a node-most-expressive GNN is not guaranteed to exist, practical GNNs/1-WL are powerful enough to discriminate almost all non-isomorphic nodes. Thus, Theorem 1 also provides insights to practical GNNs. With a practical GNN, Theorem 1 can still guarantee non-isomorphic links to be mapped to different representations as long as the practical GNN can discriminate the non-isomorphic nodes from two links (such as v2 and v3 in Figure 2). This is true almost surely according to [1].
>
> The reason why we assume a node-most-expressive GNN is because it allows a clean argument of the benefit of labeling trick (enables a GNN learning structural node representations to also learn structural link representations), and connects the concept of structural representations with GNNs. Our theorems are orthogonal to (Li et. al. NeurIPS 2020) in terms of practical significance.
>
> ##### Summarization
> To summarize, distance encoding is a special form of the labeling trick. As our first contribution, we prove that any labeling trick satisfying Definition 8 can enable a sufficiently expressive GNN to distinguish (almost) all non-isomorphic links. This is not proved in (Li et. al. NeurIPS 2020), and is raised as an open problem questioning GNN for link prediction in (Srinivasan and Ribeiro ICLR 2020). We provided a solution here, which reassures using GNNs for link prediction. Our second contribution is to define local isomorphisms to justify learning from local subgraphs. Our re-implementation of SEAL also significantly improved the state-of-the-art results of OGB link prediction datasets.
>
> [1] Laszl ´ o Babai and Ludik Kucera. Canonical labelling of graphs in linear average time. In ´ 20th Annual Symposium on Foundations of Computer Science (sfcs 1979), pp. 39–46. IEEE, 1979.

---

> > ### Comment · AnonReviewer4 · 2020-11-23
> > **Comparison with distance encoding**
> >
> > Thanks for your detailed response.
> >
> > Li et al define the distance encoding quite generally, in a manner that is very similar to the presentation in this paper. Their Definition 3.1 defines a distance encoding as a permutation invariant labelling function with dependencies on S and A, which is very similar to the definition in this paper. Their definition also includes the 0/1 "simplest" labelling as a special case. The authors do not discuss the similarities with the work of Li et al or explain how the work in this paper is complementary to Li et al, beyond stating that the DE is a special case of the labelling trick. Given the similarity between the two concepts, I would expect a greater discussion on this to demonstrate the novelty of this work.

---

### Official Review · AnonReviewer3 · 2020-10-25
**The authors discusses the difference between node embedding based method and subgraph embedding based method for link prediction tasks. However, the conclusions are not helpful.**

**Rating:** 3
**Confidence:** 5

**Review:**

The authors tried to explore the key differences between two link prediction methods. However, some statements are not precise.

The example the authors provided in the introduction is not correct. The GAE will also assign them different probabilities. When we use GAE to learn node embedding in the graphs, we usually have two different inputs: (A) nodes in the graph have input features (B) nodes in the graph do not have features and we assign each a one hot vector. For both two cases, each node in the graph in Figure 1 is unique node. Even node v2 and v3 play the same role in terms of graph structure, they still have different node embeddings considering the node features.

It is worthy to discuss that why node embeddings learning from exisiting method cannot help SEAL. It can be seen from the experimental results in two SEAL papers[1][2] that the node embedding does not help improve performance.

[1]Inductive Matrix Completion Based on Graph Neural Networks
[2]Link Prediction Based on Graph Neural Networks

---

> ### Author Response · Authors · 2020-11-15
> **It is meaningless to study the representation power of GNNs under the context of node-identifying features**
>
> We thank the reviewer for providing the context of GAE. We indeed have implicitly assumed that the graph does not have node-identifying features. However, this is a standard assumption in theoretical GNN papers. If we use node-identifying features, there is no need for studying the expressive power of GNNs, because we can always discriminate between any two nodes/links/graphs using their discriminative features.
>
> However, even there are node-identifying features, there is still a need to study the pure structure representation ability of GNNs. This is because discriminative features mean less generalizability. In the extreme case of using one-hot node features, these unique node identifiers do not generalize at all. Thus, the generalization ability of GNN still has to come from structure learning. This motivates the research in the paper. We want to not only map non-isomorphic links to different representations (which is also achievable with node-identifying features), while mapping isomorphic links to the same representation (which is not achievable with node-identifying features).
>
> By the way, the analysis in the paper allows the presentation of node features (Definition 1). That is, our theorems are applicable  to attributed graphs. However, our theorems are not applicable to one-hot node features (so are not most other theoretical GNN papers). The issue of using one-hot node features is that all the discussions on isomorphisms (Definition 3, 10, Theorem 1, 2) will make no sense, as any two nodes/links/graphs will be non-isomorphic directly due to their different node features.
>
> We agree with reviewer that it is worthy to study why node embeddings does not help SEAL, yet it is not the focus of this paper. We really hope the reviewer could reevaluate the main theorems of the paper instead of rejecting the paper only based on the example in Figure 1.
>
> We will make our assumptions on node features more clear in the revision.

---

### Public Comment · ~Guillaume_Salha1 · 2020-11-13
**Clarifications on GAE**

Dear author(s),

While I enjoyed reading the paper, I am not sure to understand the reasons why you are claiming that GAE cannot learn structural link representations. In particular, I do not understand the example from Section 1, and specifically why a GAE model should learn similar embedding vectors for nodes $v_2$ and $v_3$ from Figure 1. While I might have misunderstood something, could you please provide some clarifications on this point?

Please find below some details to corroborate my question.

------

 I fail to understand the fundamental reason why a GAE would learn similar embedding values for symmetric/isomorphic nodes, such as $v_2$ and $v_3$ in Figure 1, for the following three points.

### Point 1 - On GCN encoders

First of all, GAE models, as described by Kipf and Welling (2016), rely on GCN encoders. Assuming (as in their work) a 2-layer GCN encoder learning $d$-dimensional embedding vectors and assuming an inner-product decoder, the $n \times d$ embedding matrix $Z$ and the $n\times n$ reconstructed adjacency matrix $\hat{A}$ will be computed as follows:

$$Z = \tilde{A} \text{ReLU}(\tilde{A} X W_0) W_1 \text{ and } \hat{A} = \sigma(Z Z^T),$$

where the row $i$ of matrix $Z$ corresponds to the embedding vector $z_i$ of node $v_i$. In the above equations, $\tilde{A}$ denotes the symmetric normalization of the adjacency matrix $A$, $\sigma(\cdot)$ denotes the sigmoid function, $W_0$ and $W_1$ are weight matrices, and $X$ is a node-level feature matrix. In case of featureless graphs, as in Figure 1 of this ICLR submission, we usually simply set $X = I_n$ i.e. the $n \times n$ identity matrix, whose rows can be interpreted as one-hot vectors identifying nodes, as noted by Reviewer 3.

Therefore, although the graph structure that $v_2$ and $v_3$ "see" during message passing is identical, these nodes aggregate information from _different_ nodes (their respective neighborhood) and, in general, they will end up with different embedding vectors $z_2$ and $z_3$. This conclusion is still valid if you replace the GCN encoder from Kipf and Welling (2016) by most GNNs, such as GraphSAGE from Hamilton et al. (2017) or SGC from Wu et al. (2019).

Did you implicitly make any additional assumption on the GAE model?

### Point 2 - On the loss

Besides, I also feel that $z_1$ and $z_3$ should be different, by looking at the problem from a loss perspective. We usually tune the GCN weights of GAE models by minimizing, by gradient descent, a reconstruction loss capturing the similarity between $A$ and $\hat{A}$. In the implementation of Kipf and Welling (2016) and in most subsequent works, this loss is formulated as a cross-entropy loss:

$$L = - \sum_{i,j=1}^n [ A_{i,j} \log \hat{A_{i,j}} + ( 1- A_{i,j} ) \log ( 1 -\hat{A_{i,j}} ) ]$$

where, for any node pair $(i,j)$, we have $\hat{A}_{i,j} = \sigma(z^T_i z_j) \in  ]0,1[$.  In the example provided in Section 1, and re-discussed in Section 3.3 to justify Proposition 1, we would simultaneously have:
* $A_{2,3} = 0$, as $v_2$ and $v_3$ are _not_ connected in the graph from Figure 1;
* and  $\hat{A}_{2,3} = \sigma(z^T_2 z_3)$ very close to 1, as the paper claims that $z_2  = z_3$.

Therefore, the above cross-entropy loss would tend to… infinity. Which will hardly be a local minimum.

### Point 3 - Experiments

Last, I trained a GAE on the graph from Figure 1, using the TensorFlow implementation from Kipf and Welling (2016) with default values. I confirm that I obtain different embedding values for nodes $v_2$ and $v_3$, e.g. for a particular seed:

For $v_2$:
`[ -0.29110277  0.25207454  0.45171875  ... -2.0603905  -0.08147624 -2.7614255 ]`

For $v_3$:
`[ 1.1881522   0.9896574   0.46037757  ...  0.9059888  -1.957309    0.7554876 ]`

As a consequence, this GAE returns _different_ predicted probabilities for link $(v_1, v_2)$ and link $(v_1, v_3)$. While these probabilities vary with seeding, I observe that the returned probability for a link $(v_1, v_2)$ is always larger than for $(v_1, v_3)$ (e.g. around `0.32`and `0.04`respectively, using the above vectors). This result intuitively makes sense when looking at Figure 1.

Once again, I might have misunderstood something and/or you might have made additional (implicit) assumptions on GAE in this paper. I am looking forward to potential clarifications. Thank you very much!

---

> ### Author Response · Authors · 2020-11-15
> **Thank you for your comments.**
>
> Hi Guillaume,
>
> Thank you for your comments. We address each of your points as follows.
>
> Point 1 - On GCN encoders:
>
> As in our response to R3, we indeed have implicitly assumed that one-hot vectors are not used as node features, as this will make the discussion on graph isomorphism meaningless. Besides, using one-hot vectors would fail to map isomorphic nodes/links to the same representation. For example, in your point 3, you got different embeddings for v2 and v3. But v2 and v3 are isomorphic -- their structural representations should be the same.
>
> Point 2 - On the loss:
>
> We didn't claim z1 and z3 should be similar. v1 and v3 are not isomorphic, thus their structural representations should naturally be different. Where did you see that we claim z1 = z3?
>
> Point 3 - Experiments:
>
> Thanks for running a simulation experiment. As discussed, we don't consider one-hot vectors. If you run the same experiment with node degrees or all-one vectors as initial features, you will find that v2 and v3 will always have the same embedding.
>
>
> Please let us know if you have further questions. We appreciate your effort in understanding the paper and your questions very much!

---

> > ### Public Comment · ~Guillaume_Salha1 · 2020-11-16
> > **Clarifications on GAE**
> >
> > ### Point 1 - On GCN encoders:
> >
> > Thank you for the clarification. I would suggest mentioning this assumption explicitly in the final version of the paper, as it differs to what most GAE-related papers do in practice, notably Kipf and Welling (2016) that you cite throughout the paper.
> >
> > Besides, I am not sure to understand why the representations of isomorphic nodes "_should_ be the same" and why learning different representations (as GAE with node-level input vectors) seems to be seen as a failure in your comment. Isn't it really task dependent? For some tasks, I agree that you would want isomorphic nodes to have similar representations. But for others, such as link prediction which is the main focus of this paper, I would say that being able to learn different vectors is actually an advantage, as it directly solves the problem of Prop. 1.
> >
> >
> > ### Point 2 - On the loss:
> >
> > Sorry, there was a typo (now corrected) in my initial comment. Of course I meant $z_2 = z_3$!
> > Learning $z_2 = z_3$ while the actual $A_{2,3} = 0$ would make the cross-entropy loss of the GAE explode*.
> >
> > A workaround to this problem might be to assume another reconstruction loss for GAE with bounded reconstruction terms for each node pair (maybe the Frobenius norm of $A - \hat{A}$?). But, again, such assumption would differ to what is done in most recent GAE papers, typically those building upon Kipf and Welling (2016), and I would therefore suggest mentioning it explicitly.
> >
> > _*unless all representations tend to the 0 vector (and assuming there is no regularization to counter it), but from a learning point of view, this option will not be more satisfying._
> >
> >
> > ### Overall comment:
> >
> > Thank you once again for your time and for your detailed answers. I repeat that I enjoyed reading the paper ; my comments mostly relate to the problem's motivation and link to GAE, but absolutely not to the relevance of the proposed approach nor the quality of your experiments.

---

> > > ### Author Response · Authors · 2020-11-16
> > > **Response**
> > >
> > > #### Point 1 - On GCN encoders:
> > >
> > > Sure! We will definitely clarify this point in the revision.
> > >
> > > For why the representations of isomorphic nodes "should be the same", note that we are claiming that their "structural representations" should be the same. Structural representations are defined to be the same for structurally symmetric (isomorphic) nodes/links. But we indeed agree that sometimes it is beneficial to learn different representations for isomorphic nodes. However, these positional embeddings (such as GAE with one-hot node features, DeepWalk, and MF), as we discussed in the response to R3, have little generalizability. Although they can discriminate link (v1, v2) from link (v1, v3), they will mostly fail to map isomorphic links, such as (v1, v2) and (v4, v3), to the same representation (we let v4 be the bottom right node in Figure 1symmetric to v1). We argue that such generalizability is the main reason why SEAL can outperform node2vec, MF, etc. in the experiments. We also experimented with GAE using one-hot features, and we found except ogbl-ddi, SEAL still outperformed GAE by large margins.
> > >
> > > #### Point 2 - On the loss:
> > >
> > > Thanks for the clarification. We don't think in such case the CE loss will explode. The large deviation of $A_{2,3}$ and $\hat{A}_{2,3}$ will make $z_2^T z_3$ small. Although $z_2 = z_3$, their inner product doesn't need to be large because we have the GCN weights to turn down their scale. If it is easy, you may also run your simulation again without one-hot node features and see whether the loss will explode.
> > >
> > > #### Overall
> > >
> > > Hope these will resolve your questions. Thanks again for your interest in our paper!

---

### Comment · ~Muhan_Zhang1 · 2022-01-14
**Paper published at NeurIPS 2021.**

Our revised paper is published at NeurIPS 2021 and retitled as "Labeling Trick: A Theory of Using Graph Neural Networks for Multi-Node Representation Learning". Paper link: https://openreview.net/forum?id=Hcr9mgBG6ds. Code link: https://github.com/facebookresearch/SEAL_OGB. Thanks!

---

### Decision · Program_Chairs · 2021-01-07
**Final Decision**

**Decision:**

Reject

**Comment:**

The authors study the expressive power of Graph Neural Network architectures for the link prediction problem and provides theoretical justification for the strong performance of SEAL on link prediction benchmarks. However, The reviewers think the paper needs to improve in several aspects before it can be published: 1. More clearly explain the theoretical analysis and contribution. 2. Extensive and in-depth discussion of the similarities with and difference to the work of Li et al. to show the novelty of current work.